# Developing a Data Driven Strategy and Guideline to Increase Per Capita Open Space and Relative Accessibility in Chittagong City

Maharina Jafrin

School of Architecture & Built Environment, Deakin University, Geelong, VIC 3220, Australia; mjafrin@deakin.edu.au

**Abstract:** The population density in Chittagong City Corporation (CCC) area was 242.28 per square meter in 2019, and Bulmer suggests that, due to the high birth rate in Asia, cities such as Chittagong can be considered high density. Contextually, this 'high-density' element is a determining factor that potentially allows one to address the city's open space standard, which "should compensate and complement the physical and social context of the [urban] surrounding environment". The research in this paper is focused on the urban setting, defined in the CCC area of 168 square kilometres. The literature review and case study analysis found that per capita open space in Chittagong is far lower than the WHO recommendation (nine square meters per person). Additionally, the UN stated that "47% of [the city's] population live within 400 m walking distance to open public spaces", whereas, according to the previous study, in Chittagong City only 19% of residents live within this distance. Observing these issues, the aim of the paper is to develop an innovative way to obtain per capita open space in Chittagong city. To achieve the aim, the researchers analysed the data from surveys and interviews conducted by using SPSS and NVivo. These tools produced data that were, for example, used to develop themes of open space in Chittagong. This investigation and analysis of material allowed for the generation of strategies and planning recommendations to improve the open space situation in the city. Beyond these strategies, the research team produced new insights to promote sustainability in this area.

**Keywords:** open space; sustainability; SPSS; NVivo; survey and interview analysis

## 1. Introduction

The population density in Chittagong City Corporation (CCC) area was 242.28 per square meter in 2019 [1] (p. 20), and Bulmer [2] suggests that, due to the high birth rate in Asia, cities such as the Chittagong can be considered high density. Contextually this 'high-density' element is a determining factor that potentially allows one to address the city's open space standard, which "should compensate and complement the physical and social context of the [urban] surrounding environment" [3] (p. 27). The research in this paper is focused on the urban setting, defined in the CCC area of 168 square kilometers. The literature review and case study analysis found that per capita open space in Chittagong is 0.18 square meter [4] which is far lower than the WHO recommendation (9 square meter per person) [5]. Additionally, the UN stated that "47% of [the city's] population live within 400 m walking distance to open public spaces" [6]. Whereas, according to the previous study, only 19% of residents in Chittagong city live within this distance [4]. Observing these issues, the aim of the paper is to investigate guidelines to mitigate the ratio and to explore users' requirements of open space. The projected guidelines and the user's requirement will help to mitigate the crisis by increasing per capita open space in Chittagong city.

Therefore, the research objective is "to investigate approaches of realizing open spaces in Chittagong city" and its intent is to investigate a process to increase open

space in Chittagong city. To achieve this objective, the following two research questions were formulated:

1.　What are the city's open space aspirations and how do these meet the urban growth plan of Chittagong?
2.　What are the city's design/planning-based strategies that best support the open space aspirations of Chittagong?

After careful literature review on open space typology, strategy, and standards (published in the previous article), the researcher intended to consider local influences. In addition to provide insight, interview offered advantage in generation of research data [7] and "Surveys are used to estimate the characteristics, behaviors, or opinions of particular populations" [7] (p. 2). Therefore, to address the questions, the researcher conducted a survey and interviewed residents and professionals in Chittagong, respectively. The survey method reported by Lal (2018) [8] is considered to be appropriate for use according to this approach (with amendment) as it also revolves around a case specific social survey. In summary, in relation to this thesis, the intent of the surveys and interviews are to:

- Document resident's perceptions on existing open space quality and provision in Chittagong city.
- Document professional built environment practitioner perceptions of the existing open space quality and provision in Chittagong city.

This paper presents the analyses of survey and interview data through SPSS and NVivo. The analysis of the results influences the researcher's consideration of creating open space, indicating the types and the user's aspiration with probable guidelines to increase these spaces in Chittagong.

## 2. Interview Analysis Methodology

"[Q]ualitative interviewing, especially the in-depth interview, is now used extensively as a keyway of exploring social meaning within social science research" [9] (p. 85). The researcher interviewed 13 professionals engaged in the planning and development of Chittagong City, such as the town planner, architect, sociologist, and archaeologist. To translate the interview data into findings for discussion, the researcher implemented an inductive content analysis and thematic analysis of interview data. Thematic analysis refers to the "method of identifying, analysing and reporting patterns (or themes) within data" [10] (p. 79). In addition to a conventional paper-based approach to analysis interview data, the researcher adopted the use of the NVivo 12 software as a qualitative data analysis tool to aid in the thematic analysis of data. Following the framework of analysis stated by Braun and Clarke [10], the data collected from the interviews is examined in six phases. These are:

### 2.1. Phase 1: Familiarization with Data

The first step is to familiarize oneself with data through listening and taking notes [11]. Therefore, in this step, to facilitate the analysis, the researcher transcribed the recorded audio conversation of 13 interviews into electronic text. The interview files consist of four to 14 pages. The researcher read and then after a couple of months reread these transcripts to develop an initial view of the potential themes. In addition, for the reader's review, two interviews documented in the Bengali language have been translated into English.

### 2.2. Phase 2: Generating Initial Codes

In the second phase, for analysing the interviews, the researcher set to generate codes from the interview data. It is the researcher's consideration that "[a] code in qualitative inquiry is most often a word or short phrase that symbolically assign a summative, salient, essence-capturing, and/or evocating attribute for a portion of language–based or visual data" [12] (p. 3). To implement this consideration, the researcher prepared a coding table based on the first impression of the transcriptions, developed in phase I. To generate a

code with NVivo, the researcher imported the 13 interview files in to "NVivo". Following these, the NVivo software generated 34 codes from the 980 text references of the 13 different interview files. Accommodating the process of this large data set in this analysis chapter has been included as part of the phases' explained in this section of the thesis.

### 2.3. Phase 3: Searching for Themes

Coding of the respondents' data occurred in the first two phases, which are preliminary steps that lead to the positioning of these into groups of thematic coherence. In this sense, "[a] theme captures something important about the data in relation to the research question" [10] (p. 82) and, in this step, the researcher established the themes based on the four interview questions developed to achieve the objective. Furthermore, themes were identified based on whether thematic coherence was considered, by the researcher, as to have a semantic or latent association. That is, semantic codes and themes identify the explicit and surface meanings of the data and latent codes capture underlying ideas, patterns, and assumptions [13]. Additionally, codes can lead to the identification of interesting information in data. Themes, however, are broader and involve the active interpretation of the codes and the data [11].

The relationship between files, themes, codes (also known as nodes), and references can be simplified as in a diagram presented in Figure 1. Following this, 980 code extracts from 13 files are sorted into 3 themes.

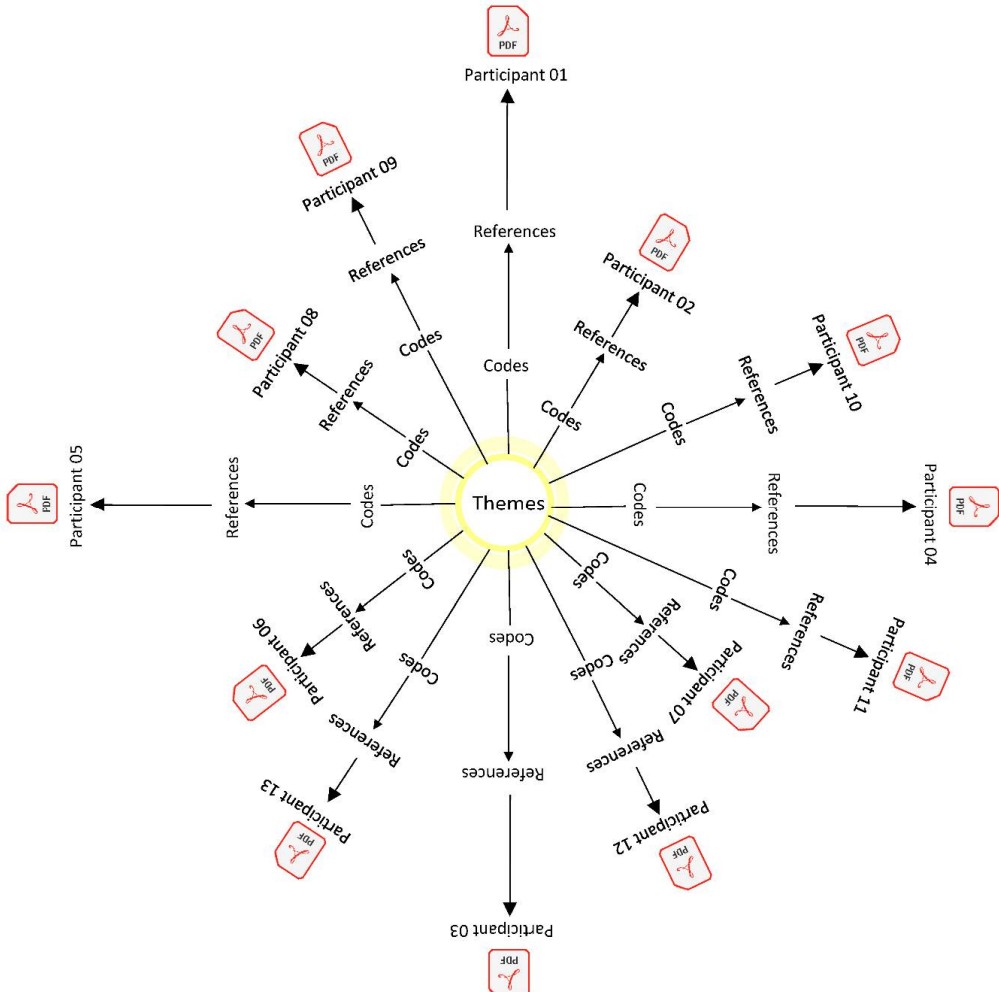

**Figure 1.** Structure of elements in thematic analysis associated with interview files derived from NVivo modified by author.

### 2.4. Phase 4: Reviewing Themes

In this step, the researcher verified "if the themes work in relation to the coded extracts and the entire data set [to] generated a thematic 'map' of the analysis" [10] (p. 87). Note, as a preliminary step, that the researcher was initially guided by a traditional paper-based approach where comments from each interview transcript were cut out and thematically arranged by code. The researcher reviewed and refined the themes identified in phase three by reading the references of each code to explore whether they support, contradict, and overlap with a respective theme [11]. In this process, the researcher sorted the references generated by NVivo into three themes. Here the references are the quotes from of interview files, and NVivo sorted the number of references into codes. NVivo was also able to derive positive and negative sentiment in the interview files. In this sense, the "Positive sentiment" indicates the probability to increase open space in Chittagong and "Negative sentiment" the contrary. For example, a code from an interview file reads as the quotation "It will carry water to the river Karnaphuli, main River and it will have beautiful, lush green banks where people can sit. It will have trees and plants, bushes and shrubs". Hence, when analysed in NVivo, this sentiment is interpreted as positive. The relationship between themes and sentiments derived from NVivo is shown in the Figure 2. The two parent sentiment nodes (Positive and Negative) have four child nodes: neutral, positive, negative, and mixed. Figure 2 shows the ratio of positive and negative sentiment with child nodes in each theme according to NVivo. This helps the researcher to understand the sentiments in each theme. Again, the sentiment ratio of positive, mixed, neutral and negative are derived from NVivo to justify the probability of increasing the open space ratio in Chittagong. This figure shows that each theme has the majority of neutral and positive sentimentit where negative and mixed sentiments are less prioritized. This means that the majority of the participants are neutral regarding open space in Chittagong, but their positive thoughts on open space in Chittagong is stronger than negative. Therefore, the themes are more positive leaning in terms of sentiment which support a respondent's willingness to create or act on the improvement of open space in Chittagong.

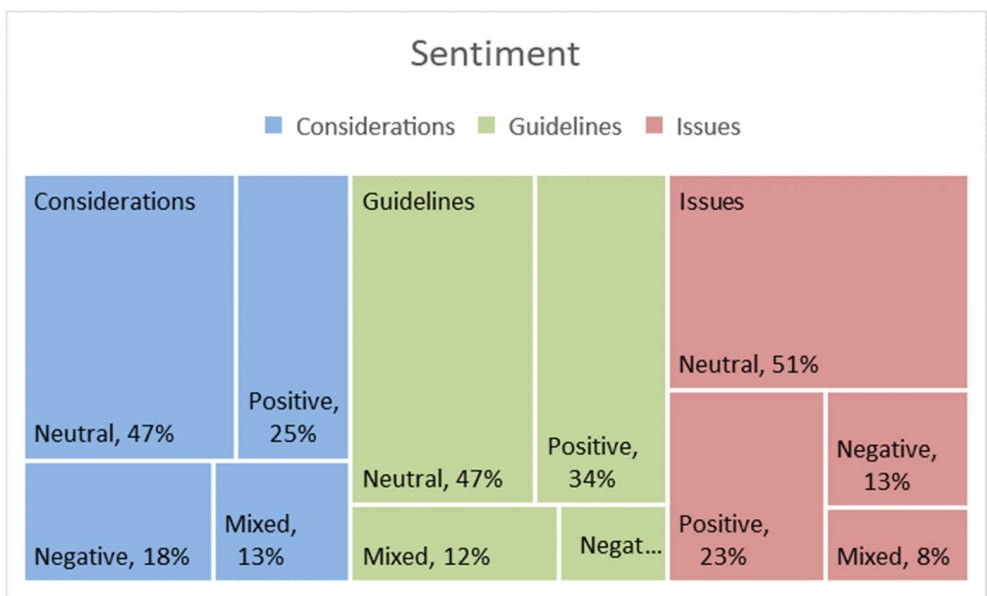

**Figure 2.** Diagram of sentiment in relation to themes.

### 2.5. Phase 5: Defining and Naming Themes

As a complementary step to phases III and iv, this phase revolves around an analysis which enhances the data placed under each theme. Hence, to generate clear titles of the themes in this step, each theme was named by analysing the overall data supporting it, [10] and this naming validation continued in the final report [13]. Following this line of theming

the data, the researcher shortlisted the codes and their associated extracts, and collated and combined the categories into broader themes. This process was proceeded by the researcher, who removed repetitions and irrelevant themes by reorganising codes and splitting differences. In addition to the NVivo result, the researcher has to add the missing nodes or rename the codes according to the findings of traditionally approached analysis.

*2.6. Phase 6: Producing Interview Report*

This phase relates the analysis of data respective to the theme to produce an analysis of the interview data [14]. It is also the final opportunity for the researchers to select an intense and persuasive set of extracts and illustrations to support the analysis [14]. When focusing on the research question, the researcher classified the codes under themes, as summarised in Table 1. After careful assessment, the researcher finalized 30 nodes with 584 references. Thirty nodes are categorised into three themes. The following sections present a more detailed data analysis of the themes and nodes.

**Table 1.** Categorization of themes and nodes.

| Themes | A. Issues | B. Considerations | C. Guidelines: |
|---|---|---|---|
| Nodes | 1. Urbanization | 1. Hydrology | 1. Typology |
| | | | 2. Waterfront |
| | | | 3. Canals |
| | 2. Incompatible land use | 2. O & M (Operation and Management) | 4. Hill development strategy |
| | 3. Calamity | 3. Accessibility | 5. Mass Transit |
| | 4. Professional body | 4. Historic landscape | 6. Railway land |
| | 5. Concurrent development | 5. Master plan | 7. Port land |
| | | | 8. Conservation |
| | 6. Land unavailability | 6. Biodiversity | 9. Small pocket land |
| | 7. Planning initiatives | 7. Climate | 10. Waterbodies |
| | 8. Incoordination | 8. Tradition | 11. Political maneuvering |
| | 9. Relocation | 9. Budget | 12. Civic engagement |

1.    Interview report:

NVivo shows that 13 interview files have 217, 208 and 185 references for considerations, or identified issues, and guidelines, respectively. Additionally, according to the NVivo, 12 people leaned towards 467 positive sentiments and 13 people interviewed lean towards 292 negative sentiments regarding open space in Chittagong. These themes helped the researcher with considerations and issues related to the development of open space guidelines while the sentiment identified to the researchers the signs of willingness that may be present to improve open space in Chittagong. Figure 3 presents the relationship between files and reference data derived from NVivo.

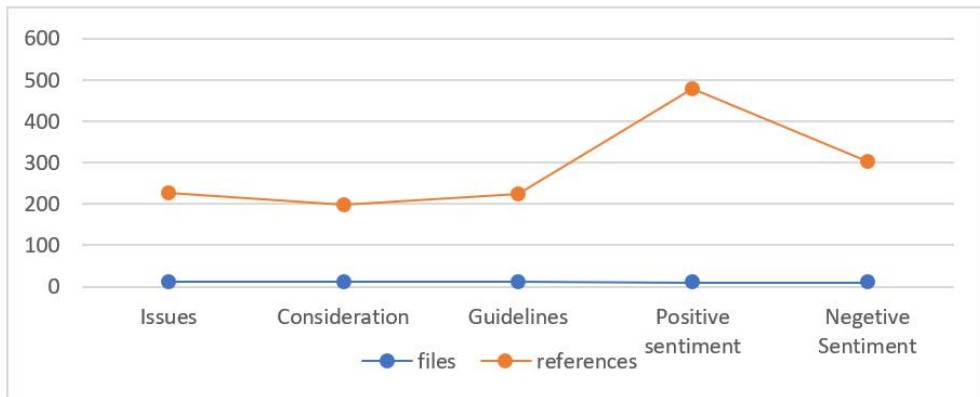

**Figure 3.** Relationship between files and reference derived from NVivo.

Hence, in this analysis a theme was developed to best position the thoughts and conceptual orientation of professionals interviewed. Three overarching themes have been identified from the interview questions, and are as follows:

1.  **Guidelines:** In response to the first interview question, "what guidelines/policies/framework do you and/or your Department/Practice have in place to address open space in Chittagong?" The researcher formulates "Guidelines" as one of the themes of the interview files. It approaches the guidelines in planning and developing open space in the city.

2.  **Issues:** This theme "Issue" was formulated from the second interview question, which was "Please tell me the issues, as you see them, relevant to existing and planned open space in Chittagong?" The theme indicates the issues related to open space in Chittagong.

3.  **Considerations:** This theme was derived from the third interview question, "Please explain to me what you/your organization is doing or plan to do to address the issue of open space in Chittagong?" This theme relates to the respondent's consideration for creating or providing open space and related facilities in Chittagong.

The fourth question of the interview was a supplementary question to the second interview question of "Can you please explain to me why you/your department doesn't consider open space as an issue in Chittagong?" grouped with the 2nd theme mentioned above.

A.  **Issues**

The interview responses presented a series of open space issues related to the open space setting in Chittagong city. Under this theme, the interviewees described the causes for reducing open space. This theme is further delineated into sub-themes where discussions on open space challenges in Chittagong are presented. These are urbanization, incomputable land use, calamity, lack of professional body, concurrent development, land unavailability, lack of planning initiative, uncoordination, and relocation. The graphical relationship (or positioning) of nodes with respect to its frequency of mention (i.e., references) are presented in Figure 4. The graph shows that "urbanization" is mostly referred to by interviewees. In addition, "Incompatible land use" and "Concurrent Development" are referred to by five respondents, represents more than one third of the total. The following section discuss the findings of the issues elaborating responses used to establish thematic framing (referred to as nodes), and the lateral references are used by respondents to describe the node.

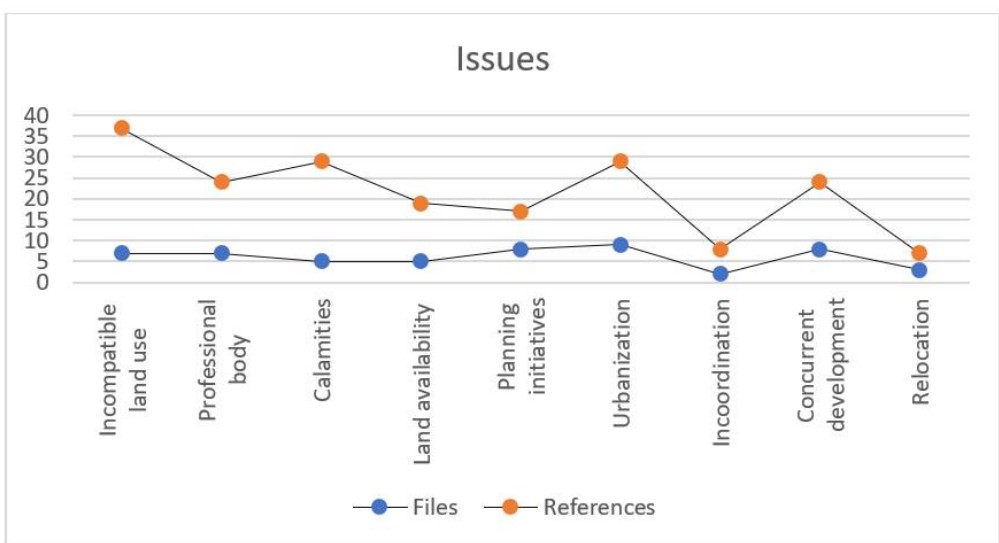

**Figure 4.** Relationship of file and references representing issues.

B.　**Considerations**

Consideration delineates the thoughts reflecting to overcome the challenges of open space. Among the thoughts, landscape, accessibility, O & M (Operation & Management), climate, master plan, tradition and hydrology were focused. Hydrology is significantly prioritised as a consideration for open space, which includes the sewage system, rainwater and tidal water management and water supply. The graphical relationship of considerations among the files and references created from NVivo is presented in Figure 5. The figure shows that hydrology is referred to 80 times by the interviewees and is the highest among them.

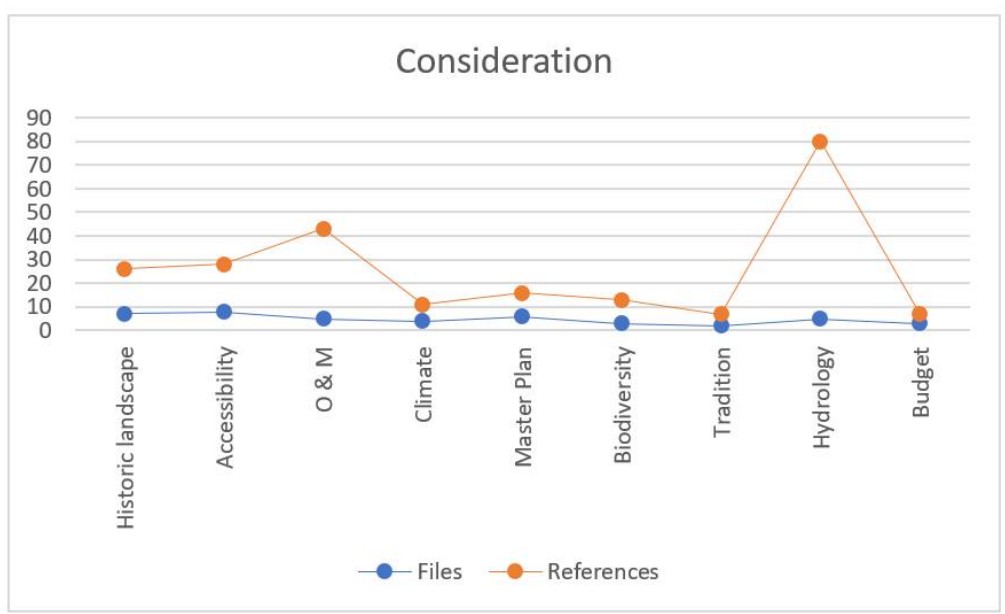

**Figure 5.** Relationship between files and reference in terms of consideration.

C.　**Guidelines**

Guidelines describes the respective measures in existing conditions to overcome the challenge of open space in Chittagong includes potentiality in dense setting. In this theme, mostly guidelines formed for existing natural reserves. Among these, canal, hills, river, ponds and the sea are recommended. In addition, guidelines regarding civic engagement,

typology and mass transit are notable. Figure 6 shows the relationship of guidelines in terms of files and references representing that hills, waterfronts, and canals are highly recommended, with guidelines as open space.

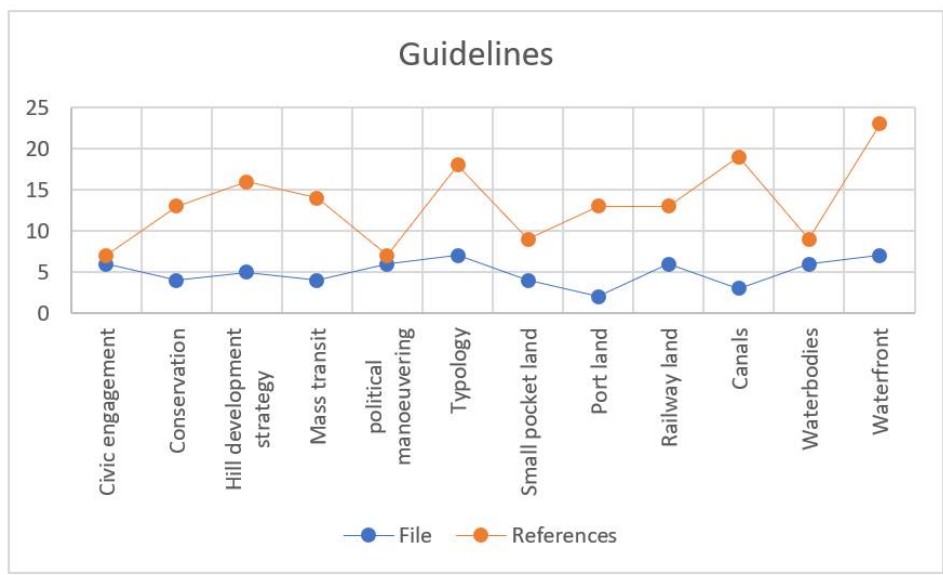

**Figure 6.** Relationship of guidelines in terms of interview files and references.

### 2.7. Findings of Interview

According to the City of Casey, the "[o]pen Space Strategy [ . . . ] provides a framework to guide the planning, design, development and management of open space" [15] (p. 2). The interview files uncover thoughts on planning and opportunities of open space in Chittagong City. Barth [16] (p. 35) stated that, "[c]areful and thoughtful planning is critical to identifying opportunities to generate greater resiliency and sustainability benefits for the community". The key findings derived from the analysis are as below:

1.   A need to set up a "Park and Open space body" which will implement the proposals on open space in the master plan, reserve the open spaces, mitigate jurisdictional conflict, maintain an open space policy, execute management, maintenance, land acquisition and preservation by hiring a number of professionals such as a planner, sociologist, economist etc. Where hiring a group of professionals will be cost effective, engaging them by outsourcing from professional bodies and social organizations like IAB (Institute of Architects Bangladesh) and FPC (Forum for Planned Chittagong), who are willing to work, investigate, survey and publish newsletters and articles on open space.

2.   The current practice of open spaces like Jamboree Park, Patenga beach, and Lal Deghi needs to be revisited and can be encouraged by the development of CRB, DC Hill, and Parade Ground for their distinctive, aesthetically pleasing and significant recreational aspect.

3.   Planning initiatives need to be taken for open space policy and setting open space standards and hierarchies in open space development. Open space development should be initiated in the smallest areas such as "Ward" in the Chittagong corporation area. While preparing open space policy hierarchy from smallest to largest administrative area it is suggested to follow by 41 wards to 13 police stations in the city. It is necessary to segment the planning in each ward according to action for 1 year, 5 year and 10 years etc.

4.   Due to jurisdictional conflict, co-ordination between 23 organizations working for Chittagong city need to be set up for conserving open spaces such as seashores, riverbanks, hills and canal sides.

5. Proposals on open space in the Master Plan need to be executed. These are: the 1995 Master Plan's direction to conservation of hills and use of the riverbank and seashore as open space, the 1961 Master Plan proposals on 26 open spaces in and around the city, and the guideline of DAP 2015 to create Chittagong Park and Recreation Department are significant [17] (pp. 2–10).

6. Due to the scarcity of land, waterfront, ponds, canals, riverbank, seashore, and hills, these are recommended to be preserved as open space in Chittagong City. Additionally, there is a need to propose a land bank in the future extension proposal of the city.

7. Natural open spaces in Chittagong such as hills, waterbodies and waterfronts need to be accessible. Figure 7 shows the relationship of open spaces according to reference the guidelines on these are as follows:

    I.      A need to preserve natural open space such as hills, seaside, open space and low-laying areas in densely urban setting resistingnatural calamities like hill slide, cyclone, earthquake, and flooding.

    II.     A need to preserve open space for silt trap, flood storage pond and rainwater reservoir.

    III.    In addition, to access the hills, respondents encouraged recreational development on hilltops with a maximum 10% land coverage. To connect neighbourhoods with open space connectivity and accessibility through walkways, removing blockages on the footpath and the establishing of mass communication systems needs to be undertaken.

    IV.    The relocation of settlements from the hills, rivers and the canal side of the city during eviction needs to be executed to use existing open space and their accessibility needs to be improved.

    V.     There is a need to create a considered hydrological system in the city. Open space will increase the rainwater catchment area and recharge ground water. Low-laying areas need to be preserved to avoid flash flooding. The canal's capacity for discharging water into the Karnaphuli River needs to be increased by removing the encroached settlement. A dam needs to be created in the hills to increase the rainwater catchment area, as this will help to increase water supply and to produce electricity. A separated sewerage system needs to be developed to use the existing 200 miles of canals as an environmentally friendly open space corridor.

    VI.    Sea front and river development along the ring road (15 miles in length of the Karnaphuly River and approximately 11 miles length in Bay of Bengal) are proposed in the transportation Master Plan, and can provide multiple types of open space either with free access or with ticket entry to attract all income groups, and any other type of developments are strictly discouraged in this area.

    VII.   Fifty-seven canals in Chittagong can contribute to open space as linear park. The proposal of a silt (sediment) trap by CWASA can reduce sedimentation in the canal system and serve as open space. The sides of the canals are 200 miles in length and can provide multiple benefits such as the free flow of water, lavish greens with seating, walkways, bike tracks, and promote ecology and biodiversity along the water channel.

    VIII.  The continual hill range in the north from Foy's Lake to Sitakunda are proposed to develop by setting water treatment plant (desalinization of sea water) and rainwater harvesting system with dam in hilly areas to produce electricity and captivate water for future use for mitigating crisis of supply water will promote recreational open space and reduce flooding. Only 10% development is encouraged in order to promote limited (residential, health or educational) land use with a meandering road to increase accessibility.

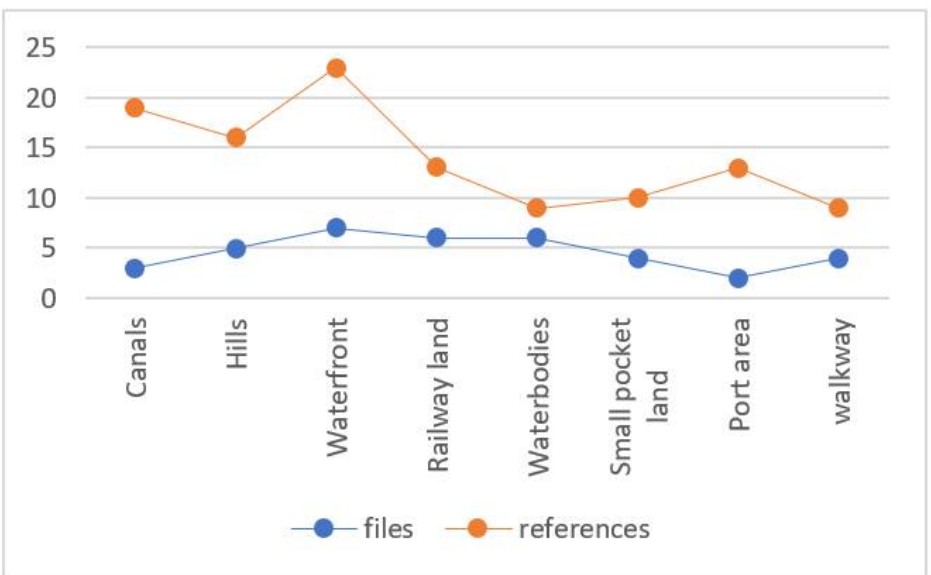

**Figure 7.** Probability of open space resources.

8.  A need to install an operation and management system. The management cost can be mitigated for supervising, managing regulations, and repairing damage by engaging stakeholders who are benefitted can provide informal and passive supervision. A need to increase surveillance by users to minimise risk of anti-social activity in open space setting. Maintenance, quality control such as cleaning, lighting system, watering and dewatering during monsoon need to be considered. Open space needs to have free accessibility and a control free environment.

9.  Historic playgrounds, open fields, and waterbodies in Chittagong need to be preserved to contribute as open spaces. Boundaries of historic open space need to be demarcated. These types of places include a polo ground, CRB, Batali Hill, Ashker Deghi, Lal Deghi and Foy's Lake for their natural landscapes.

10. To create open space, biodiversity needs to be considered by planting trees for seasonal fruits and trees for habitats and the hatching of butterflies.

11. There is a need to consider local climate in creating open space. Chittagong has a tropical monsoon climate. Respondents suggested that in Chittagong the open spaces need more shading, and shelter will benefit the users from the hot climate and heavy rainfall.

12. There is a need to consider the traditional use and riverine culture of Chittagong in open space planning. While considering festivals in winter for harvesting crops, organising local fairs with traditional "Boli Khela" (traditional wrestling) is exemplified. The tradition of celebrating "Pohela Boishakh" (Bengali new year) requires large open space that can only can be accommodated in CRB and DC Hill. Another strong open space requirement is "Eidgah" for religious prayer that is held twice a year. In addition, a mosque needs open space to accommodate additional devotees once per week. The respondents directed that these open spaces are lost due to the extension of prayer halls and the inclusion of commercial use. They strongly recommend keeping open space in front of the mosques and to preserve the specific open space for Eid prayer in front of Jamatul Flah Mosque and specific open space "Laldighi Maidan" for "Boli Khela".

13. A need for adequate government funding and new source of revenue for open space development. Government's needs to allocate significant budget in open space development.

14. There is a need to investigate the quality and quantity of open space in Chittagong and set up the mixed used of open space settings such as park and playground arrangements. Open space needs to be used vibrantly in the daytime and night-time

to cope with the surroundings without destroying its quality. Open space should have an arrangement for all types of users, such as playgrounds for the young, parks for all aged users, and walking trails for the elderly and other ancillary facilities like adjacent tea stalls or coffee shops and libraries. The interviewees refer to create more footpath in this city, providing shelter, resting point and seating, walking or bicycle track, shadings with plants and trees along the footpath and open spaces.

15. A need to Install mass transit that will help to connect open spaces in the city and serve the residents who are deprived of outdoor recreational facilities close to their home. Mainly CRB hills, Foy's Lake and Batali Hills, Jorr Deghi (pond), Bhelur Deghi (pond), Debar Par (pond) in metropolitan area or CBD (Central Business District) referred as prominent open spaces in the city owned by Bangladesh Railway. The interviewees suggested that the city's huge railway land, preserved for future use, can be integrated as open space, a bazaar (informal shops), which would help to connect the city and provide open space while transporting goods and passengers. The land on both sides of the rail line connecting Chittagong with north-south and with Chittagong Port can contribute to linear parks to create green corridors connecting open spaces in Chittagong.

16. Chittagong has five beautiful waterbodies, and these waterbodies need to be demarcated and protected as open space, including Debar Par, Jorr Deghi, Asker Deghi, Foy's Lake and Bhelur Deghi. Providing accessibility in these settings can contribute to open space in Chittagong.

17. There is a need to locate small pocket lands in Chittagong along the road or without any access. This type of land will be cost effective and serve as open space in dense settings. The government can acquire these lands by paying compensation to create parks for neighbourhood. These small open spaces in some cases can open up large chunks of land that could benefit the governmentto initiate large scale projects.

18. Policy makers need to take initiative to develop open spaces such as Jamboree Park and DC hill park development convey by political commitment and willingness.

19. Citizens need be aware of the open space crisis in Chittagong and open space standards practiced around the world. There is also a need to conduct a survey of citizens to know the requirement and type of open space. They need to be aware of cleanliness and engaged in management and maintenance. Social workers, politicians, technical personnel, and historians are strongly recommended to engaged in maintenance and the management of these areas.

## 3. Survey Analysis Methodology

A survey of the city's open spaces is an appropriate tool to address the second question of the objective, because it can lead to estimates on population characteristics [10] and the potential demand of these open spaces. The target group of this research is park playground and open space users. Only 95,000 people live in proximity to open spaces. The target group of this research is park playground and open space users. The researcher applied "convenience sampling" directed by [18]. It involves the researcher selecting participants simply for reasons such as ease of access, in terms of physical proximity and accessibility [18]. Again, Bryman [19] (p. 97), suggested that "[a]s the sample size increase, sampling error decrease". To decrease sampling error, a minimum size of a sample has been selected from a daily user's ratio. The average daily users in CRB, Jamboree Park and Parade Ground are 1421, 2570 and 1285 people, respectively (see Table 2). Therefore, the number of survey respondents that have been selected from daily users is minimum of 100 per site. In total, three hundred respondents participated in the questionnaire survey.

**Table 2.** Findings of comparative analysis of demographic survey in three sites.

| | | Jamboree Park | CRB | Parade Ground |
|---|---|---|---|---|
| **Gender** | | Male participants are slightly higher than female participants; ratio is 56:44 | Male participants are moderately higher than female participants; ratio is 69:21 | Male participants are dramatically higher than female participants; ratio is 87:13 |
| **Age group** | | Highest number of participants are in age group of 25–35 | Highest number of participants are in age group 35–45 | Highest number of participants are in age group 18–25 |
| **Education** | | Most of the participants are graduates. | Most of the participants are graduates. | Most of the participants are educated up to the higher secondary level. |
| **Profession** | | Participants are mostly students, work in the business sector and arehousewives/dependents. | Most of the participants are housewife/dependents. | Most of the participants are students. |

To address this issue, park, playground and open space users were targeted for the survey due to their catchment area in Chittagong City. Among the seven types of open spaces in Chittagong [10], the survey has been conducted in three types defined as publicly accessible open spaces. Each sample site selected distinctively exemplifies a respective type of open space selected for the survey due to their recent design changes and upgrading, which attracts more users. These open space, park and playground developments are remarkable in Chittagong, because the preservation and development of existing open space did not come into practice until 2015. However, only one public park, one playground and one open space have been developed since 2015 in Chittagong. There are numerous other private open spaces and amusement parks in Chittagong that have been developed or transformed to ticketed entry which have not been considered. The three sample sites in this thesis, however, were selected, as they were solely developed as public open spaces, accessible to the city's residents free of charge and have seen an increase in the number of regular users. The researcher therefore has selected these open spaces for their contemporary importance to the city's residents. In this research, selected sites are known as the Parade Ground, Jamboree Park and Chittagong Railway Building (CRB) area representing sample of the city's playground, city park and open space respectively. A survey on three types of open space in Chittagong has been conducted to perceive the user's perception and demand on open space. A total of 279 participants responded to the survey. The survey data is analyzed with SPSS (Statistical Package for the Social Sciences) as below. The analysis is described following a presentation of the structure of the questionnaire, such as the 1st tire will describe the demographic information, the 2nd tire will explain the user's response to the development of existing open spaces, the 3rd tire will present the scenario of available park playground and open space in neighbourhoods and the 4th tire will demonstrate the availability of natural open space in close proximity to the users.

*3.1. 1st Tier: General Information*

Data presented in this section is derived from the first level or leveled as "A" in the questionnaire and is designed to provide an overall understanding of the demographics of open spaces users. That is, the data on gender, age group, education, and occupation of the participants respective to each study site. Figure 8 shows the overall graph on the general demographic information of the respondents presented in the series of questions A1 to A5. The data shows that among 279 respondents, 73% were male and 27% were female. This male dominated response rate is interesting because compared to female users, male users are predominantly higher in number (finding is presented in the fifth tier). The data shows

that, among the respondents, the age group between 18–25 represented 47% of the total, the age group between 25–35 represented 18% of the total, and the age group between 35–45 represented 18% of the total. This highlights that most respondents are young. Looking at the education level of the participants, the survey found that, 40% of participants are graduates and 40% have completed their education to the secondary level. This is a general statistic on the type of users in open spaces. In terms of occupation, the result presents that 36% of respondents were students, 28% of respondents were housewives or dependents, 12% of participants were businesspeople, and the rest were engaged in other professions. In conclusion, the information points out that respondents are mostly male, young students and that most of the female respondents were housewives.

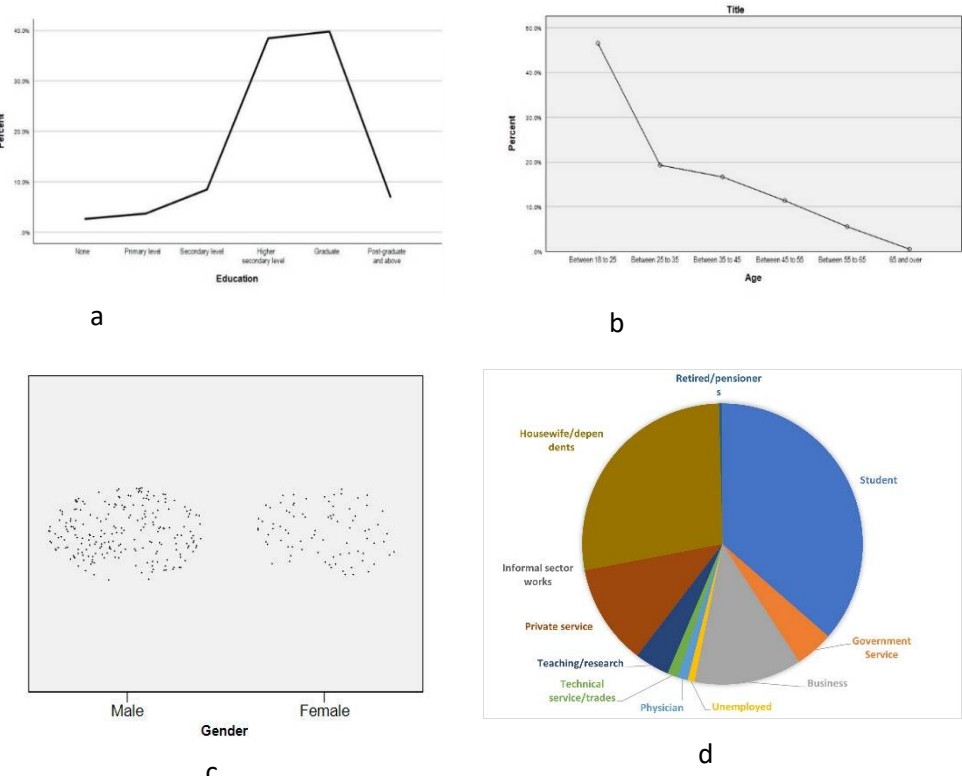

**Figure 8.** Demographic survey of education (**a**), age (**b**), gender (**c**), and occupation (**d**).

In terms of the comparative analysis of participants in three sites, Figure 9 indicates that in Parade Ground, 83 male participants were the highest in number. In contrast, Jamboree Park had the highest female participants, and they represented a total number of 41 among 100 participants. Among the six age groups, 61 participants aged 18–25 are predominantly participated in Parade Ground survey. In Parade Ground, 61 participants are educated up to higher secondary level. Among the occupation groups in Parade Ground, students are the highest number of respondents. In CRB, 52 female participants were housewives and dependents, which is the highest among the three sites.

From SPSS analysis, the researcher produced the following table for comparative analysis of the three sites (Table 2).

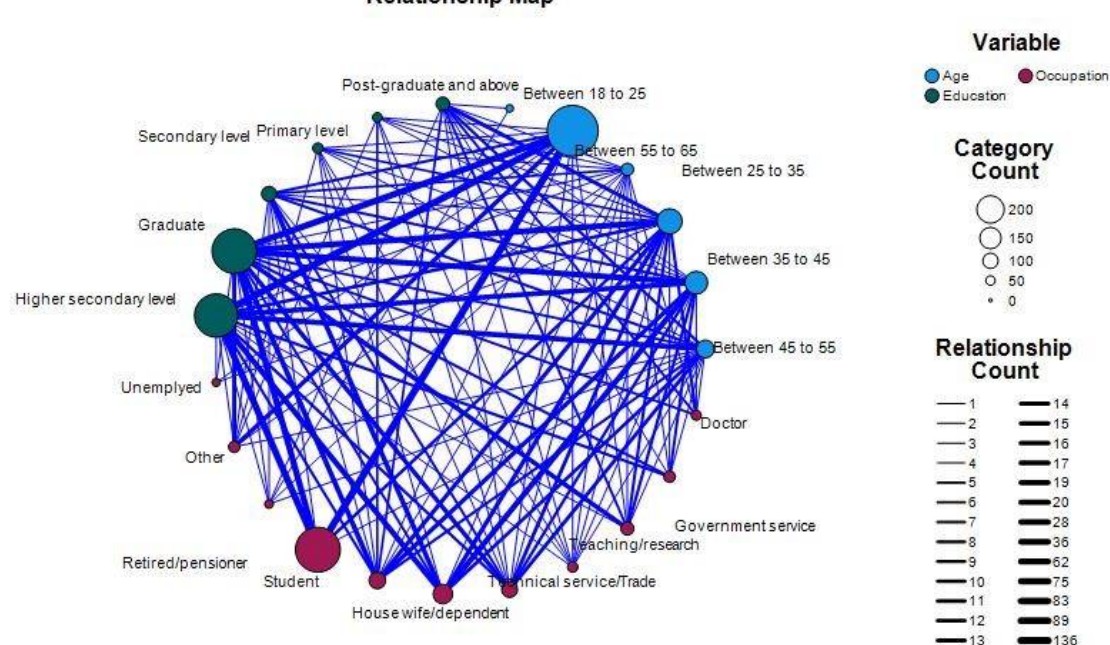

**Figure 9.** Correlation of Age, occupation, and education status in three sites derived from SPSS.

*3.2. 2nd Tier: Park, Playground and Open Space User*

This tire is formated with data gathered in the second section of the questionnaire to investigate the user's interest to use open space and whether their demands are mitigated by these settings. The second section of the questionnaire, considered as level "B" in the questionnaire set, is targeted at examining demand, accessibility, use of the formal open space setting and how the open space's development has influenced its use.

For better understanding by grouping relative questions from the second section of the questionnaire, the findings are divided into four sections. The first section assessed the frequency of the visit before and after an open space development to understand the resident's demand on the city's open space. The next section of the questionnaire is focused on gathering data on the user's accessibility of the open spaces. It concentrates on user's travelling distance, mode of transportation and pedestrian accessibility facility. The third part of this section concentrated on the type of use, their attitude and aspiration towards activities in the open space and the purpose of their open space visit. At the end of this section, "development influence" is considered to measure whether the user's aspiration is meeting by the facilities and quality provided in open space. The following figures of this section are summarized from the survey outcome from questions presented. The analysis of different parts of this tier can be described as follows.

a.   **Demand of open space:** This section derives the visiting frequency of users in three sites to determine the demand of open space. Figure 10 shows that, cumulatively, the user's daily and weekly visiting ratio are 30% and 35%, respectively. The survey identifies that Jamboree Park and Parade Ground has the highest visiting frequency, and no substantial differences in daily and weekly users' ratio, but CRB has more weekly visitors compared to daily visitors. This survey data also shows that more than 80% of visitors surveyed stated that they cannot make time to visit more frequently. In addition, more weekly visitors in CRB shows that this place serves as a city park, while more daily visitors in Jamboree Park and Parade Ground shows that it serves as a local park and playground, respectively. According to a survey, 52% of Parade Ground users think that the space is not enough for them. The users stated that to use the playground, they have to come first before it is occupied by others. The fact that users have to wait to play illustrates that they need more playgrounds. 32% of

Parade Ground users do not have walkways along the street connecting their homes to the playground.

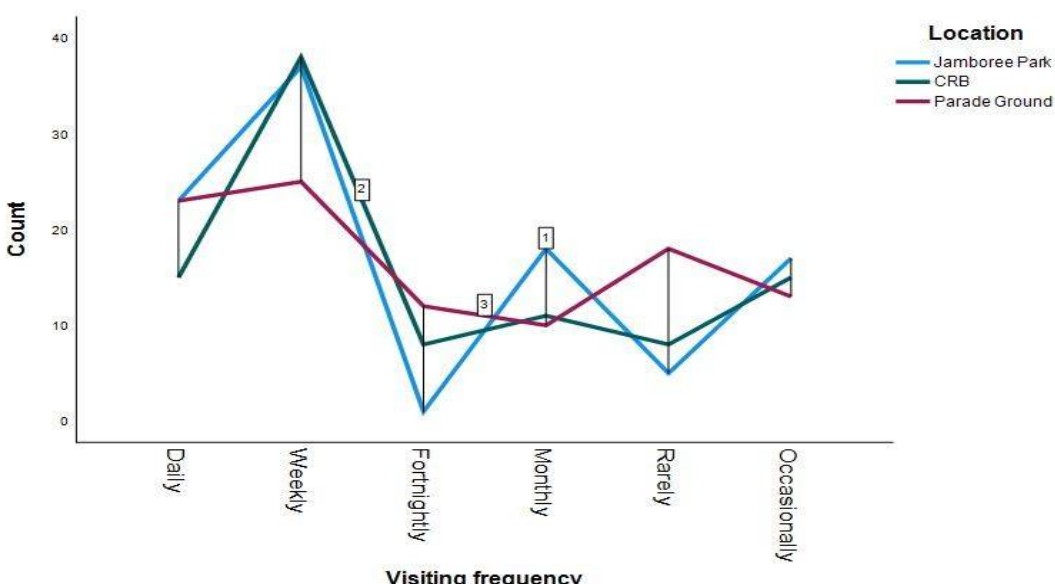

**Figure 10.** User's visiting frequency.

b.  **Travelling distance:** The cumulative analysis of three sites in SPSS shows that a total of 32% of users commute less than one kilometer in distance, 51% of users commute from 1–5 km distance, 11% of users visit 5–10 km distance and only 0.08% of visits more than 10-km in distance to get into these open space settings. The data indicates that a majority of users travel 1–5 km distance to get into the open spaces. Singly, CRB has 16% of users from 0–1 km distance, 61% of users from 1–5 km distance and 14% of users from 5 km to 10 km distance. This data indicates that CRB has more distant visitors compared to neighbourhood visitors. On the other hand, Jamboree Park has 43% of users from 0–1 km distant and 45% of users from 1–5 km distance. The Parade Ground has 39% users from 0–1 km distance and 47% of users from 1–5 km distance and 12% of users from 5 km to 10 km distance. Therefore, both the Parade Ground and Jamboree Park has mostly visitors from 0–1 km distance and 1–5 km distance, but CRB visitors from 1 to 5 km distance are prominent than 0–5 km distant visitor. Figure 11 shows the travelling distance of visitors in each park.

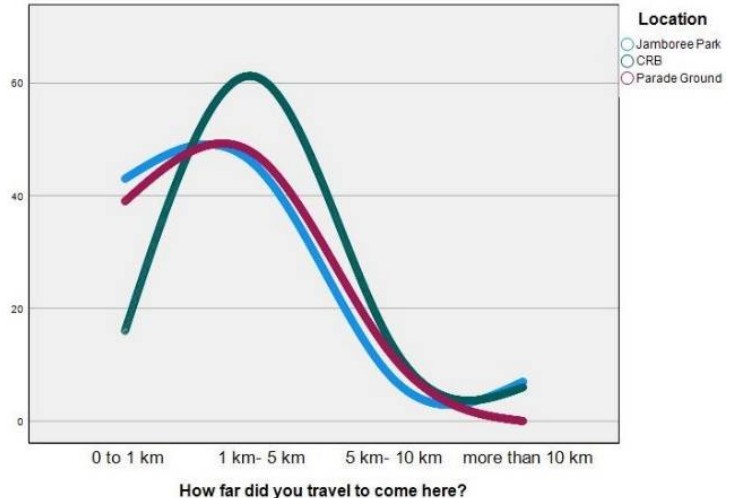

**Figure 11.** Travelling distance of user.

c.   **Mode of transportation:** In Jamboree Park and Parade Ground, 56% and 57% of visitors walk to the open space, while in CRB, only 16% of users walk to get into the place. Most of the CRB visitors ride either a bus or rickshaw (a light two-wheeled passenger vehicle manually pulled by one person carries two passengers at a time and mainly used in Asian countries) to travel Ito the place. This data complies with the travelling distance of distant CRB users as discussed above. Figure 12 shows the comparative analysis of the mode of transportation in three sites. In summary, the Jamboree Park and the Parade Ground has more neighbourhood visitors compared to distant visitors. The survey did not find any visitor in the Parade Ground commute that came from more than 10 km in distance.

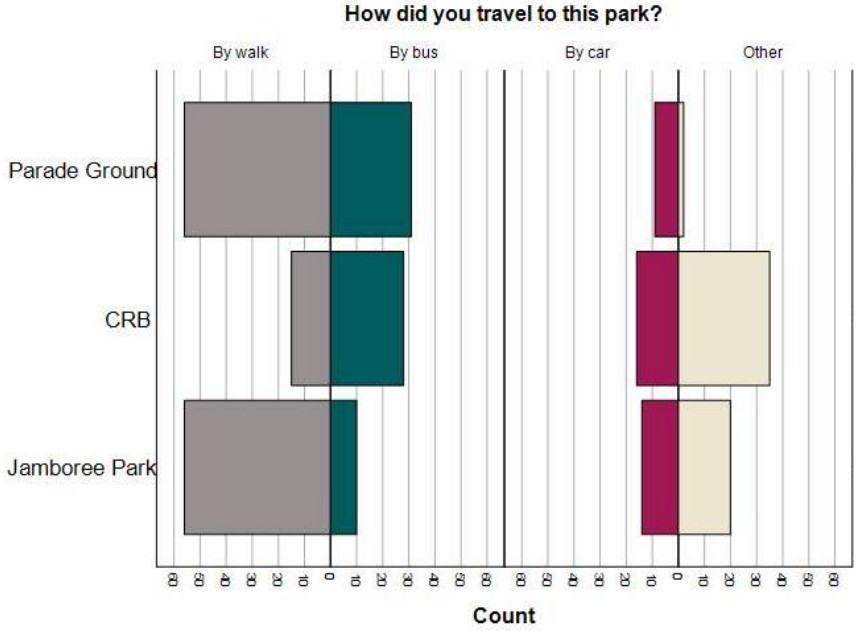

**Figure 12.** Mode of Transportation.

The data comply with the mode of transportation, as most of the visitors in Jamboree Park and Parade lives up to 5-km distance and generally walks to these places. Distant CRB users ride the bus, car and rickshaw to get to the destination.

d.   **Influence of transformation:** This part is to investigate whether development of open space setting influenced the user's visiting frequency. The survey shows that development of Jamboree Park, CRB and Parade Ground influenced the regular users to increase their visiting frequency by 90%, 48% and 38% respectively. Again, 72% of the Jamboree Park visitors, 41% of the CRB visitors and 33% of the Parade Ground visitors started to visit the open space settings after development. The result indicates the development has affected the users visiting frequency.

The fact that three percent of Jamboree Park users, 15% of CRB users and 19% of Parade Ground user were not satisfied with the development shows that most of the users appreciate the developments in the three sites. Figure 13 represents the comparative analysis of the influence of the development of the open spaces to the users. It indicates that 4% of Jamboree Park users, 48% of CRB users and 55% of Parade Ground users' visiting frequency after development remains unchanged. In addition, the visitors were asked for the reason of their increase and decrease of visit in the open space settings. It was an open-ended question, and the answers are sorted in Table 3. In response to this question, the survey found that 40% of users of the Parade Ground claimed that the field is not sufficient and can't accommodate all users. In addition to this, the accessibility restriction (only adjoining college students can use it) in Parade Ground was also notified.

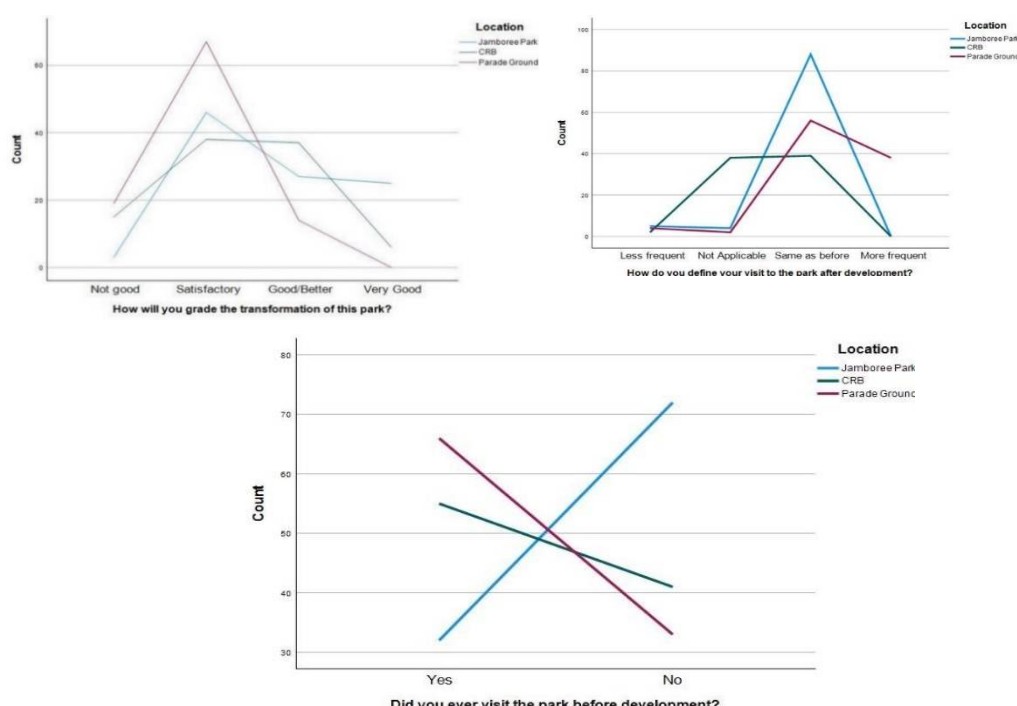

**Figure 13.** Influence of transformation scaling by increase and decrease of visit before and after transformation.

**Table 3.** Findings for increase and decrease of visit in each site. JP-Jamboree Park, CRB-Central Railway Building, PG- Parage Ground.

| Reason for Increase | JP | CRB | PG | Reason for Decrease | JP | CRB | PG |
|---|---|---|---|---|---|---|---|
| Good environment | √ | | | Lack of maintenance | | | √ |
| Better weather protection | √ | | | Not enough space | | | √ |
| Green space | | √ | | Flooding | | | √ |
| Calm/quite/peaceful | | √ | | Bad environment | | | √ |
| Park development | √ | | | Restricted accessibility | | | √ |
| Safety and security | √ | | | | | | |
| Evening lighting | √ | | | | | | |
| Openness | | √ | | | | | |

　　　In summary, the survey finds that these developments predominantly increase the user's visit and satisfies them. Among the three sites, the Jamboree Park development increased user's visitation frequency compared to CRB and Parade Ground. Furthermore, the survey shows that safety and security in Jamboree Park influence users to increase their visit to the park. The visitors strongly recommend that the Parade Ground cannot accommodate all the users and they need to wait for their turn to play.

e.　**Purpose of visit:** To answer the question, respondents were asked to choose the activities they are most likely to do in the open space settings, and around 33% stated that they visit open spaces to enjoy with their family and friends, 10% visit for sightseeing and 15% for walking. In Parade Ground, more than 56% visit for playing and 16% visit for watching matches. However, 70% of female users of Parade Ground declared that they use the field for walking and jogging, 20% of visitors watch matches and 10% of the users play. In summary, Parade Ground is predominantly used by

male users for playing and infrequently used by female users, mostly for walking. Jamboree Park and CRB users are kin to visit for recreation and socializing.

*3.3. 3rd Tire: Neighborhood Park, Playground, and Open Space*

This section is designed to gather data on the availability and demand of open space in Chittagong. To investigate this query, the visitors of each open space were asked whether they have available formal open space in their neighborhood or not, whether they think it is sufficient or not, and in case of park and playground arrangement, which one they prefer most. The survey result shows the scenario of the user's behavioral impact on the neighborhood park use explained as follows:

a. **Availability of Neighborhood Park, playground and open space:** This section was designed to gain data on user's availability and demand of theneighborhood park, playground and open space. The responses identify that 16% of users have a park, 33% of users have a playground and less than 2% of users have both a park and playground in their neighborhood. The result indicates the necessity of open space in Chittagong. In addition, 47% of users neither have a park nor a playground in their neighborhood. Among them more than 11% of users want to have a park, more 10% of users want to have a playground in their neighborhood and 79% of users want to have both (park and playground).

b. **Purpose of open space:** To answer this question, the respondents were asked to choose the purposes they would like to visit in desired parks and open space. Nearly one-quarter of the respondents said that they would want open space for recreation, 20% of respondents want it for its openness, more than 20% of users prefer it for social interaction and 15% of users want it for exercising.

c. **Park and playground arrangement preference:** When respondents were asked to choose among the park and playground, most of them picked combined arrangements of park and playground. The data indicates that, 11% of users think that they should have a park in their neighborhood, 10% of users think that they should have a playground in their neighborhood and 79% of users thinks that they should have both park and playground in their neighborhood. In the Parade Ground 86% of users think that they should have a playground like this in their neighborhood for kids up to 15 years of age. According to the survey, 95% of Jamboree Park users think that they should have more parks like this. The result reveals that users desire more parks and playgrounds in Chittagong city.

d. **Open space availability for kids:** Most of the parents visiting the open spaces stated that their kids stay at home and do not play outdoors. The survey shows that less than 32% of parents can send their kids to the playground and more than 65% parents can't send their kids to the playground as it is unavailable in their neighborhood. The question discloses shortness of playground for kids in Chittagong city. In summary, most of the residents do not have parks or playgrounds in their neighbourhood. This section was followed by asking the respondent how frequently they visit open spaces close to their neighbourhood and how far is the open space setting from their home. The result shows that more than 70% respondents pick "not applicable" as they do not have this setting. In addition, this shows that playground for kids are not available.

e. **Open space close to workplace/study:** The survey shows that 45% of users have open space close to their workplace and 45% of users do not have open space close to their workplace/study area. More weekly visitors in CRB shows that this place serves as city park, while more daily visitor in Jamboree Park and Parade Ground shows it serves as local park and playground, respectively.

*3.4. 4th Tier: Natural Open Space and Its Accessibility*

As Chittagong City has a range of natural open spaces like hills, creeks, sea and river, this section of the questionnaire was designed to generate data whether the residents

have natural open spaces close to their neighborhood and if these places are accessible to them. Here accessibility is coined as access through walkways or roads without blockage or control. Figure 14 shows that 61% of users have natural open space close to their neighborhood. Among them 27% have a canal or creek nearby, 11% of user have pond, 9% of users have hills, 8% of have sea beach and 6% of users have river close to their neighborhood. 39% of users declared that they do not have any natural open space close to their neighborhood. Again, among the 61% of users who have natural open space in their neighborhood, 55% of users do not have accessibility to the natural open space, where accessibility in terms of physical connectivity has been discussed in previous chapter. Seventy-three percent of users who do not have accessibility to natural open spaces close to their neighborhood stated that if they are willing to visit those places. 44% of users stated that they do not like visit natural open space close to their neighborhood as the places are not clean and safe.

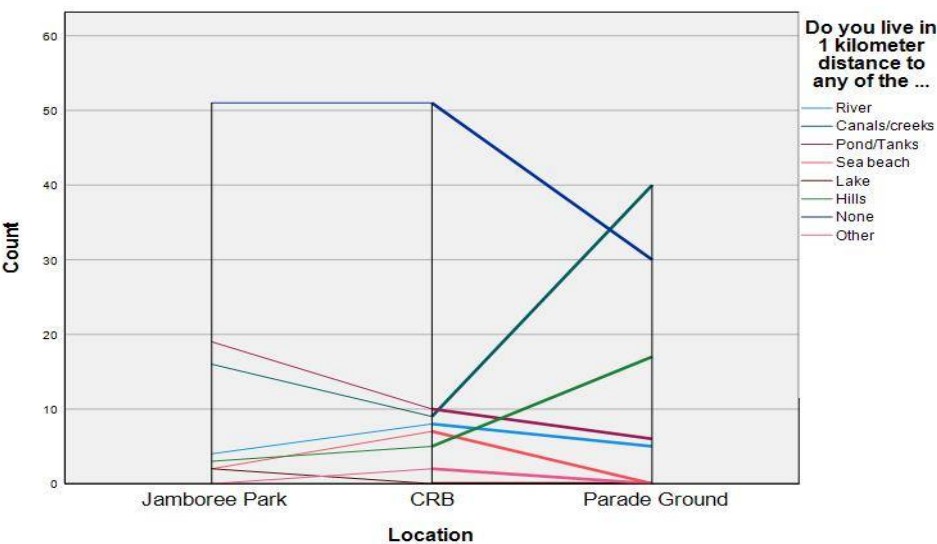

**Figure 14.** Availability of natural open space close to neighbourhood.

In summary, nearly 60% of users could have open space proximity to their home, if the natural open spaces in Chittagong City are accessible, clean, safe and well-maintained. The survey indicates that natural open spaces have more probability to contribute to open space in Chittagong City.

*3.5. 5th Tier: Relativeness of the Park, Play Ground and Open Space*

To get an intense data, the sites were surveyed both on weekday and weekend. At the end of the survey, a relative evaluation has been done among these three sites. To perceive the user's ratio of open space, total amount of users and its relation to the area was analyzed as follows:

a.  **Number of users:** To compare the number of users among the open spaces, users per square meter has been considered. From the demographic survey, it shows that among the two city parks, the Jamboree Park is the most populous. In Jamboree Park, 3000 users visited the park in weekend from 5:00 p.m. and 6:00 p.m. Compared to park size, 11. 85 square meter per person area occupied the park at this pick time. According to Lancaster R.A. [8] (p. 70), a neighborhood playground (3–5 acres) has 264-person capacity and a community recreation center (10–15 acre) has 420–820-person capacity. Compared to this data, Jamboree Park holds 4 times the visitors than the standards recommended. The park is incompetence to holds the highest number of users at pick time (5–6 pm), which can be illustrated through the Table 4. Again, Parade Ground holds 850 users at its pick time (5–6 pm) and CRB has the least number of visitors compare to Jamboree Park (CRB area is partially open for visitors). The

Table demonstrate the data on entry, exit and total users calculated during the survey. The following table illustrated the ranking of users and area of the park, playground and open spaces surveyed in this research. The survey shows that, in weekend 4 p.m. to 7 p.m. is the pick time in these open space settings to hold the highest visitor.

**Table 4.** Area distribution per visitors in Jamboree Park, CRB and Parade Ground.

| Open Space | Area | Day | Total Visitors | Opening Hours | Average Visitors/h | Visitors in Pick Hour (5–6 p.m.) |
|---|---|---|---|---|---|---|
| Jamboree Park | 8.55 acres/ 35,550 sqm | Weekday | 3140 | 9 | 348 | 1250 |
| | | Weekend | 4750 | 9 | 527 | 3000 |
| CRB | 12 acres/ 50,000 sqm | Weekday | 573 | 12 | 31 | 185 |
| | | Weekend | 2270 | 12 | 126 | 950 |
| Parade Ground | 6.11 acres/ 25,000 sqm | Weekday | 1180 | 7 | 131 | 750 |
| | | Weekend | 1390 | 7 | 154 | 850 |

From Table 4, the user's area ratio can be expressed in the following graph (Figure 15). Calculating daytime as active hour in CRB and opening hour in Jamboree Park, the survey indicates that Jamboree Park has the highest number of average visitors.

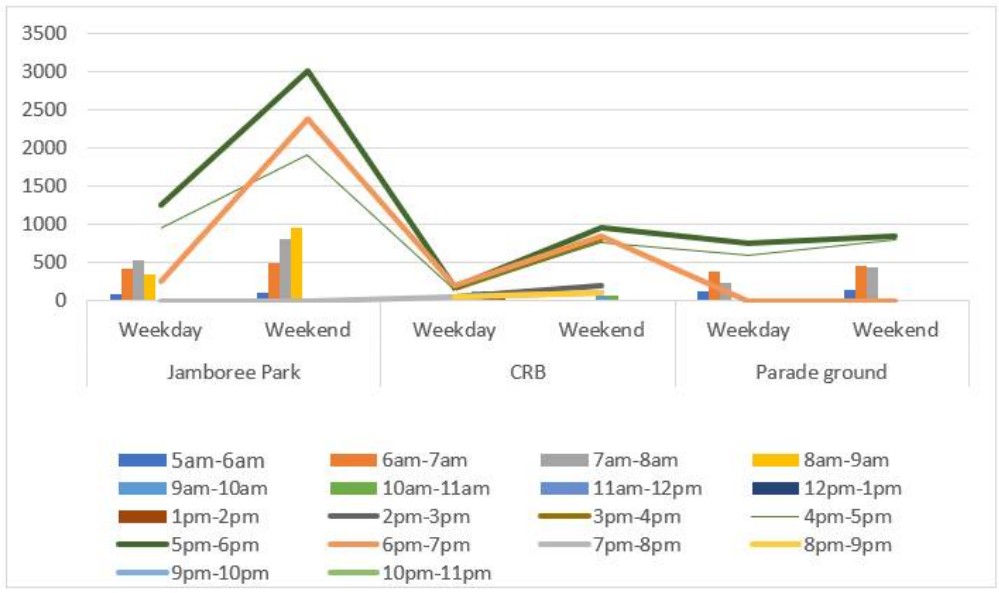

**Figure 15.** Comparative analysis of number of users.

b. **Male and female user's ratio:** This section is designed to get comparative data on open space and user ratio of the three sites surveyed in this research. The male/female ratio in Figure 16 identifies that among the three sites Jamboree Park has more and Parade Ground has less female user and vice versa. The survey shows that, in weekdays, female users decrease and male users increase. In summary, women are more likely to visit on the weekend and least likely to visit on weekdays. In contrast, the male user ratio increased in weekdays. The overall result shows that the female ratio in the Parade Ground is least. While asking the reason for more and less use of the open spaces, female users of Jamboree Park stated that they feel more secure here and female users of Parade Ground stated that they only visit the playground for walking. The result shows that among the three sites, on the weekend Parade Ground and Jamboree Park's average visitors per hour is 197 and 154 persons, respectively. According to the open space standard the number of visitors per square

meter indicates that the open spaces are overcrowded. The result indicates the demand of open space in Chittagong City. Among the three sites, Jamboree Park is the most popular open space due to its safety and maintenance. The Parade Ground is the second most popular and CRB is the least popular open space.

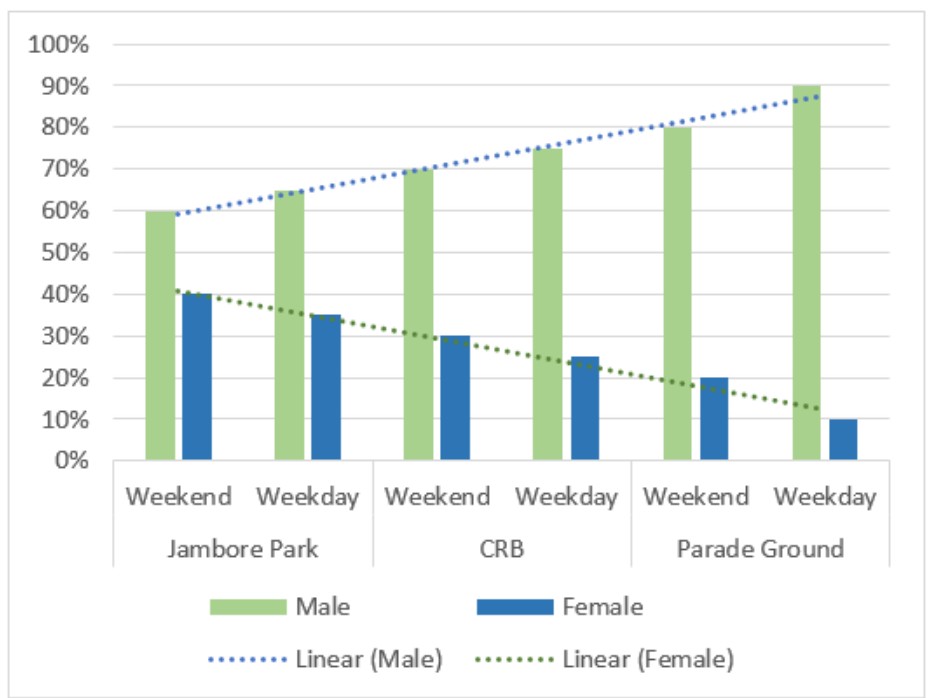

**Figure 16.** Male and female user ratio.

*3.6. Findings of Survey*

From the analysis above the following findings are mentioned as below:

1. More than 80% visitors surveyed stated that they cannot make time to visit more frequently.
2. Fifty two percent Parade Ground users think that the space is not enough for them. The users stated that to use the playground, they have to come before it has been occupied by others.
3. Thirty two percent Parade Ground users do not have walkways along the street connecting their home to the playground.
4. Thirty two percent of users commute less than one kilometer distance, 51% of users commute from 1–5 km distance, 11% of users visit 5–10 km distance and only 0.08% of users visits more than 10-km distance to get into these open space settings.
5. CRB has more distant visitors compared to Jamboree Park and Parade Ground. Most of the users of Jamboree Park (88%) and CRB (86%) live within a 5 km radius.
6. In Jamboree Park and Parade Ground, 56% and 57% of visitors walk to the open spaces, while in CRB, only 16% of users walk to get into the place.
7. Development of Jamboree Park, CRB and Parade Ground influences the regular users to increase their visiting frequency by 90%, 48% and 38%, respectively. Again, 72% of Jamboree Park visitors, 41% of CRB visitors and 33% of Parade Ground visitors started to visit the open space settings after development.
8. That three percent of Jamboree Park users, 15% of CRB users and 19% of Parade Ground users were not satisfied with the development shows that most of the users appreciate the developments in the three sites.
9. Forty percent of the users of Parade Ground claimed that the field is not sufficient and cannot accommodate all users.

10. Around 33% of users stated that they visit open spaces to enjoy with their family and friends, 10% of users visit for sightseeing and 15% for walking. In Parade Ground, more than 56% of users visit for playing and 16% of users visit for watching matches. However, 70% female users of Parade Ground declared that they use the field for walking and jogging, 20% of visitors watch matches and 10% of users play.
11. 16% of users have a park, 33% of users have a playground and less than 2% of users have both a park and playground in their neighborhood. In addition, 47% of users neither have a park nor a playground in their neighborhood.
12. More than 11% of users want to have a park, more 10% of users want to have a playground in their neighborhood, and 79% of users want to have both (i.e., park and playground).
13. Forty-five percent of users want open space for recreation, 20% of respondents want it for its openness, more than 20% of users prefer it for social interaction and 15% of users want it for exercising.
14. 86% of Parade Ground users think that they should have a playground like this in their neighborhood for kids up to 15 years of age. 95% of Jamboree Park users think that they should have more parks like this.
15. More than 65% of parents can't send their kids to the playground as it is unavailable in their neighborhood.
16. Sixty-one percent of users have natural open spaces close to their neighborhood. Among them 27% of users have a canal or creek nearby, 11% of users have pond, 9% of users have hills, 8% of users have sea beaches and 6% of users have rivers close to their neighborhood.
17. Fifty five percent of users do not have accessibility to the natural open space, where accessibility in terms of physical connectivity has been discussed in the previous chapter. Seventy-three percent of users who do not have accessibility to the natural open space located close to their neighborhood stated that they are willing to visit those places. 44% of users stated that they do not like visit natural open spaces close to their neighborhood, as the places are not clean and safe.
18. In Jamboree Park, 3000 users visited the park on the weekend from 5:00 p.m. to 6:00 p.m. and per person area is11. 85 square meter at this time.
19. Jamboree Park accommodates four times more visitors than the standards recommended [16] (p. 70).
20. Parade Ground holds 850 users at its chosen time (5:00 p.m. to 6:00 p.m.) and CRB has the least number of visitors compared to its designated open space area (the CRB area is partially open for visitors).
21. In weekend 4:00 p.m. to 7:00 p.m. is the mostly chosen time in these open space settings that hold the highest visitor.
22. Jamboree Park holds the highest number of average visitors among the open space settings.
23. Female users of Jamboree Park stated that they feel more secure here and female users of Parade Ground stated that they only visit the playground for walking.
24. Among the three sites, in weekend Parade Ground and Jamboree Park's average visitors per hour is 197 and 154 persons, respectively. According to open space standard, the number of visitors per square meter indicates that the open spaces are overcrowded.
25. Jamboree Park is the most popular open space due to its safety and maintenance. Parade Ground is second most popular and CRB is the least popular open space.
26. The survey shows that 45% of users have open space close to their workplace and 45% users do not have open space close to their workplace/study area.

## 4. Discussion

The findings of the interview and the survey individually presented coincide in some aspects. The survey highlights residents' interest to use open space and how the development of the park and playground inspire them to be more active. It also emphasized

the limitation of these spaces in cases of security and cleanliness. On the other hand, the interview specifies the potentiality of existing open spaces. It is guided to use city's railway land, hills and riverside and canals by mixed use (park with playground) development that will not only serve commuting to places, installing power generating and drainage system but also promote potential open space in the dense setting. Thus this spaces will also benefit the city environmentally and physically. The enlightment on waterlogging derived from interview coincide with survey survey result by user's limitation to use the park during monsoon. The proposal of hill development that arose in interviews coincides with the security problem in open space found in the survey. Furthermore, the proposal of mass transit will increase distant users in open spaces like CRB. Again, the consideration of cultural distinction emphasized in the interview reflects the female users dissatisfaction of using Parade Ground. Table 5 is an example of findings that can be use as feedback to survey which can be extended in the design section. Individually, the survey reflects the users' interest to use open space and expectation to existing open spaces. In contrary, the interview reflects professional's thought to support the user's need of open space. It also broadens the limitations and prospects to create open space in the city.

**Table 5.** Feedback from interview reflecting on the survey.

| Supporting Question | Feedback from Interview | Feedback to Survey |
|---|---|---|
| How to create new open space? | Seperating the canal system from swarage, remove settelement on both side of canals and conserve as mandatory open space. | Increase connectivity to natural creeks, provide walking trails and promote biodiversity. |
| | Preserve low laying areas. | Increase ground water catchment area and decrease flooding while serving as open space. |
| | Develop hills with providing dams, resorts and trails in limitation of 10% ground coverage. | Promote security, connectivity and restrict landslides while significant concern has been made during development. |
| | Develop mass transit. | Encourage independence of motor vehicles, connect city parks, promote day and night time use of areas close to station. |
| How to develop existing open space? | Ensure security, connectivity, visibility and accessibility. | Encourage more users. Promote direct and indirect use of open space. |
| | Ensure peoples' participation to maintain the space. | Engage more users and create awarness among residents. |
| | Ensure proper drainage/water collection during monsoon. | Help neighbours to use water in crisis. |
| | Promote mixed use such as parks and playgrounds. | Will engage both visitors and players at a time. |

## 5. Conclusions

Among the physical components that facilitate urban development, public open space is singularly responsible for improving urban quality [20]. The objective of the survey was to examine "how public open space meets residents' need" [21] (p. 11). To promote planned open space, in addition to a literature review, the interview built the platform to create open space in the city with its limitations and the survey extend the necessity to create open space. The analysis of the interview and survey with NVivo and SPSS helps the researcher to reach the findings and creates a platform to deal with the open space

in Chittagong City. The findings of the survey explore the crucial demand of parks and playgrounds in the city. The interview explored the potential natural open spaces and their limitations for use. The findings from the interview will be used to plan the open space network by ensuring its compatibility. The findings showing the maximum opportunity in creating an open space network in Chittagong and the probability will help to increase its per capita ratio in the city. Thus, these findings sketch the plan and draw a concrete guideline and proposals to increase public open space in Chittagong city.

**Funding:** This research received no external funding.

**Institutional Review Board Statement:** The study was conducted in accordance with the "DEAKIN UNIVERSITY HUMAN RESEARCH ETHICS COMMITTEE" and approved by the Institutional Review Board (or Ethics Committee) of Deakin University "(Deakin project ID: 2019-243 and 8 October 2019)." for studies involving humans.

**Informed Consent Statement:** Written Informed consent was obtained from all participants in interview involved in the study and verbal consent has been taken from all participants in the survey. humans.

**Data Availability Statement:** The survey and interview data are preserved by Deakin University for the participant's privacy purpose. Any data relating to this study can be provided with the permission of Deakin University committing privacy control.

**Conflicts of Interest:** The author declares no conflict of interest.

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
