# Peer review of "Developing a Data Driven Strategy and Guideline to Increase Per Capita Open Space and Relative Accessibility in Chittagong City"

_sustainability, doi:10.3390/su14169828_

Round 1

Reviewer 1 Report

The subject of access to the greenery in cities is very actually. Unfortunately, the research presented in this article is chaotic.

There is no literature research on the analyzed problem.

In the introduction, the author states that "the aim of the paper is to develop an innovative way to achieve per capita open space in the Chittagong City". Unfortunately, this goal was not achieved at work.

The questionnaire surveys among experts and inhabitants are in no way correlated with each other or it is misspelled because the reader does not see them. I understand that these were to be two stages of one study, but the summary results of the study were not presented. There are separate conclusions after each stage of the survey. There is no summary covering both stages and no innovative way to achieve per capita open space in the Chittagong City.

You need to think about the number and descriptions of the charts. There are too many of them, some of them do not contribute anything and some are unfortunately illegible.

Author Response

Thank you for your inputs.

The literature research on the analysed problem was published in another article and referred in this article with publication details in the reference list. 

To correlate the survey and interview a separate paragraph named "discussion" has been provided. Due to short period of time it is limited and can be extend further.

Number and description of the report has been rearranged. To ease confusion, less prioritized figures are taken away.

Thank you.

Reviewer 2 Report

In times of the current climate crisis, open spaces are being redefined in the process of urbanization. More attention is paid to issues related to environmental protection and to issues related to solutions such as the circular economy or broadly understood water saving in cities. The goals adopted by the global policy on urban development change the traditionally understood urban planning into one that is more friendly to natural ecosystems and based on broadly understood resilience. The research results presented in the article show that this subject is noticed not only by experts in spatial planning, but also by open space users. An example of an extensive analysis presented by the author on a Chinese example shows a number of aspects that are related to the attractiveness of open spaces, such as, among others, behavioral, functional, communication and compositional elements. Considering the above, the presented study is certainly an interesting and important topic from the point of view of the contemporary debate on urbanization. However, in its present form, the presented article needs to be supplemented and rearranged.

The basic issue is the very poor preliminary discussion and, in principle, the lack of a final discussion. At the moment, the perception of contemporary urbanism is changing a lot, which causes attempts to redefine this field of knowledge in contemporary reality. Therefore, a lot of research and publications related to this topic are being produced, so the author has a very wide selection of existing studies that he could include in the research background. I consider it necessary to significantly expand this element of the article. Exactly the same situation applies to final discussion, which is missing at the moment. I believe that it is worth enriching the article with this content, because the presented research is valuable and interesting.

The structure of presenting the methodology and results would also need to be reorganized. At the moment, one with the other is mixed up and gives the impression of chaos. The methodology for such a complex study should be presented in an additional, more detailed diagram. The diagram shown in Figure 1.1. it is too simplified in relation to the presented study. Methodology and results should be clearly and clearly separated, at the moment one smoothly flows into the other, which makes it difficult to recognize the logic of the presented study. The very structure of the presentation of results also requires some sorting, perhaps a separate graphical diagram would be indicated, specifying individual groups of results and their interrelationships.

Another important issue is the numbering of Figures and Tables. The division into Figures 1 and Figures 2 groups is incomprehensible as it introduces unnecessary mess. Perhaps it would be worthwhile to use more simple numbering.

With such an extensive study, the conclusions are far from sufficient and far too general. Perhaps it is worth including general recommendations regarding open spaces, resulting directly from the study presented. Partly, such elements now appear in the part showing the results of the study, perhaps it would be worth redrafting and dividing it.

Author Response

Accordingly background of the research has been extended in the introduction. A separate discussion point with elaboration has been provided. 

The structure has been rearranged by providing separate methodology of each study. Figure 1 has been modified for further understanding. a sample table has been provided presenting interrelation of the analysis. Due to short period of notice, it is only providing focus point. 

Numbers of figures and tables has been rearranged according to the suggestion.

The conclusion has been extended with more thoughts.

Thank you, again.

Reviewer 3 Report

Dear Author,
your article presents an interesting approach to the topic and quite an extensive, multileveled research – described very interestingly  and precisely. A thorough description of the research analysis process proves the reliability of the Author. The article has a clear structure as a whole, especially the part that describes the research and conclusions from it in great detail. Nevertheless, the reviewer sees a few weighty issues that seem important and in their opinion should be addressed before the article could be submitted for publication. 

Please see below a list of suggestions/comments for additions and complimentary information  for your kind consideration:

1. in the 'Introduction' the Author indicates that the purpose of the article as such is to „investigate approaches…”, and then  that "its intent is to propose a process to increase open space in Chittagong city". Ultimately, we do not see the implementation of this goal in the article, but only a mention that "these findings will be use to draw guideline and proposals to increase public open space in the Chittagong city". It is suggested to verify the purposes of the article. Maybe it was that the research carried out by the Author had such a goal, but the article itself is to present rather only one stage of the research? Anyway this should be clear and reflected within the text;

2. in sections 2.1.3 and 2.3. such thing as an "Objective 3" appears, it is not clear what it is, since at the beginning of the article only one main objective was indicated;

3.  lack of the "Discussion" paragraph along with a critical discussion of the research and its results;

4. in the 'Conclusions' section, according to the reviewer, the conclusions should include a direct discription of the relationship between the main objective, the 2 main research questions and the results of both studies presented in the article. Some kind of analysis of the relationship between the conclusions of the NVivo study with the SPSS study as an element that would collectively prove the sense of using the selected 2 types of research methods to achieve the assumed goal – would be worth considering. For example, a table juxtaposing the findings of the interviews with the specialists versus conclusions of the questionnaire and the SPSS survey? To be considered;

5. too few references (16) in the Bibliography section, In the 'Introduction' section, it would be possible to extend the references to the issue of, for example, creating green public spaces 10-15 minutes from places of residence (etc.), more indepth contextualisation with references to other researchers' works would seem appropriate too., which would also strengthen the bibliography itself;

6. perhaps the reviewer did not catch it in the content of the article somehow, but it seems not entirely clear according to what detailed criteria the social analysis was undertaken in these three particular locations. Also, they are mentioned to have a different typology, but also not explicitly discussed. Suggestion to describe it more clearly in the article.

Author Response

Dear reviewer,

Thank you for your extensive observations. Your thoughts are implemented in the article as below:

  1. The line "its intent is to propose a process to increase open space in Chittagong city" has been modified. sorry for the confusion. As the article is presenting only one stage of the research, the purpose has been corrected.
  2.  Sorry for confusion, "objective 3" has been modified. At this stage there is only one objective. 
  3. Discussion of the findings has been added, thank you.
  4. The conclusion has been extended. Due to small time frame, a brief table has been provided in discussion section.
  5. Number of references has been increase to 21.
  6. The details of sites and the particular reason of choosing these sites has been added under survey analysis. Thank you.

Round 2

Reviewer 1 Report

I accept the article as it stands.

Reviewer 2 Report

The article has been significantly improved and is suitable for publication in its present form. The research background contained in it was initially broadened and is now sufficiently presented. The authors specified the research questions, which significantly improved the structure of the article. The introduction of a workflow for the description of materials and methods worked similarly. A very positive aspect is that the text is supplemented with a discussion. The structure of this new chapter is clear and does not raise any doubts. A positive element is the addition of a table organizing the issues that the authors referred to. I am still not fully convinced by conclusions that could be more elaborate, but at the moment this is the only element that may raise minor doubts.

Reviewer 3 Report

Dear Author/s, thank you for your effort and improvements. I do not have any more comments.